# Greedy Hash: Towards Fast Optimization for Accurate Hash Coding in CNN

**Shupeng Su**[1]     **Chao Zhang**[1*]     **Kai Han**[1,3]     **Yonghong Tian**[1,2]

[1]Key Laboratory of Machine Perception (MOE), School of EECS, Peking University
[2]National Engineering Laboratory for Video Technology, School of EECS, Peking University
[3]Huawei Noah's Ark Lab
{sushupeng, c.zhang, hankai, yhtian}@pku.edu.cn

## Abstract

To convert the input into binary code, hashing algorithm has been widely used for approximate nearest neighbor search on large-scale image sets due to its computation and storage efficiency. Deep hashing further improves the retrieval quality by combining the hash coding with deep neural network. However, a major difficulty in deep hashing lies in the discrete constraints imposed on the network output, which generally makes the optimization NP hard. In this work, we adopt the greedy principle to tackle this NP hard problem by iteratively updating the network toward the probable optimal discrete solution in each iteration. A hash coding layer is designed to implement our approach which strictly uses the sign function in forward propagation to maintain the discrete constraints, while in back propagation the gradients are transmitted intactly to the front layer to avoid the vanishing gradients. In addition to the theoretical derivation, we provide a new perspective to visualize and understand the effectiveness and efficiency of our algorithm. Experiments on benchmark datasets show that our scheme outperforms state-of-the-art hashing methods in both supervised and unsupervised tasks.

## 1   Introduction

In the era of big data, searching for the desired information has become an important topic in such a vast ocean of data. Hashing for large-scale image set retrieval [7, 8, 21, 22] has attracted extensive interest in Approximate Nearest Neighbor (ANN) search due to its computation and storage efficiency with the generated binary representation. Deep hashing further promotes the performance by learning the image representation and hash coding in the same network simultaneously [30, 15, 18, 6]. Not only the common pairwise label based methods [17, 30, 4], but also triplet [34, 35] and point-wise [19, 31] schemes have been exploited extensively.

Despite the considerable progress, it's still a difficult task to realize the real end-to-end training of deep hashing owing to the vanishing gradient problem from sign function which is appended after the output of the network to achieve binary code. To be specific, the gradient of sign function is zero for all nonzero input, and that is fatal to the neural network which uses gradient descent for training. Most of the previous works choose to first solve a relaxed problem discarding the discrete constraints (e.g., [34, 19, 31] replace sign function with tanh or sigmoid, [20, 36, 17] add a penalty term in loss function to generate feature as discrete as possible), and later in test phase apply sign function to obtain real binary code. Although capable of training the network, these relaxation schemes will introduce quantization error which basically leads to suboptimal hash code. Later HashNet [4] and Deep Supervised Discrete Hashing (DSDH) [16] made a progress on this difficulty. HashNet starts

---

[*]Corresponding author

training with a smoothed activation function $y = tanh(\beta x)$ and becomes more non-smooth by increasing $\beta$ until eventually almost behaves like the original sign function. DSDH solves the discrete hashing optimization with the discrete cyclic coordinate descend (DCC) [26] algorithm, which can keep the discrete constraint during the whole optimization process.

Although these two papers have achieved breakthroughs, there are still some problems worthy of attention. On the one hand, they need to train a lot of iterations since DSDH updates the hash code bit-by-bit, while HashNet requires to increase $\beta$ iteratively. On the other hand, DCC which is used to solve the discrete optimization by DSDH, can only be applied to the standard binary quadratic program (BQP) problem with limited applications, while HashNet is still distracted by quantization error as $\beta$ cannot increase infinitely. Therefore this paper proposes a faster and more accurate algorithm to integrate the hash coding with neural network.

Our main contributions are as follows. (1) We propose to adopt greedy algorithm for fast processing of hashing discrete optimization, and a new coding layer is designed, in which the sign function is strictly used in forward propagation against the quantization error, and later the gradients are transmitted intactly to the front layer which effectively prevents the vanishing gradients and updates all bits together. Benefitting from the high efficiency to integrate hashing with neural network, the proposed hash layer can be applied to various occasions that need binary coding. (2) We not only provide theoretical derivation, but also propose a visual perspective to understand the rationality and validity of our method based on the aggregation effect of sign function. (3) Extensive experiments show that our scheme outperforms state-of-the-art hashing methods for image retrieval in both supervised and unsupervised tasks.

## 2 Greedy hash in CNN

In this section we will detailedly introduce our method and first of all we would like to indicate some notations that would be used later. We denote the output of the last hidden layer in the original neural network with $\mathbf{H}$, which also serves as the input to our hash coding layer. $\mathbf{B}$ would be used to represent the hash code, which is exactly the output of the hash layer. In addition, $sgn()$ is the sign function which outputs +1 for positive numbers and -1 otherwise.

### 2.1 Optimizing discrete hashing with greedy principle

Firstly we put the neural network aside and focus on approaching the discrete optimization problem, which is defined as follows:

$$\min_{\mathbf{B}} \ L(\mathbf{B}),$$
$$s.t. \ \mathbf{B} \in \{-1, +1\}^{N \times K}. \tag{1}$$

$N$ means there are $N$ inputs, and $K$ means using $K$ bits to encode. Besides, $L(\mathbf{B})$ can be any loss function you need to use, e.g., Mean Square Error loss, Cross Entropy loss and so on.

If we leave out the discrete constraint $\mathbf{B} \in \{-1, +1\}^{N \times K}$, aiming to obtain the optimal continuous $\mathbf{B}$, we can calculate the gradient and use gradient descent to iteratively update $\mathbf{B}$ as follows until convergence:

$$\mathbf{B}^{t+1} = \mathbf{B}^t - lr * \frac{\partial L}{\partial \mathbf{B}^t}, \tag{2}$$

where $t$ stands for the $t$-th iteration, and $lr$ denotes the learning rate.

However, it is almost impossible for $\mathbf{B}$ calculated by Equation (2) to satisfy the requirement $\mathbf{B} \in \{-1, +1\}^{N \times K}$, and after considering the discrete constraint, the optimization (1) will become NP hard. One of the fast and effective methods to tackle NP hard problems is the greedy algorithm, which iteratively selects the best option in each iteration and ultimately reaches a nice point that is sufficiently close to the global optimal solution. If $\mathbf{B}^{t+1}$ calculated by Equation (2) is the optimal continuous solution without the discrete requirement, then applying the greedy principle, the closest discrete point to the continuous $\mathbf{B}^{t+1}$, that is $sgn(\mathbf{B}^{t+1})$, is probable to be the optimal discrete solution in each iteration thus we update $\mathbf{B}$ toward it greedily. Concretely we use the following equation to solve the optimization (1) iteratively:

$$\mathbf{B}^{t+1} = sgn( \ \mathbf{B}^t - lr * \frac{\partial L}{\partial \mathbf{B}^t} \ ). \tag{3}$$

Conceptual convergence of Equation (3) is shown on Figure 1, from which we can see that each update of our method is able to reach a lower loss point $(-1, 1) \rightarrow (-1, -1) \rightarrow (1, -1)$ and finally reach the optimal discrete solution $(1, -1)$ of the loss contour map in Figure 1.

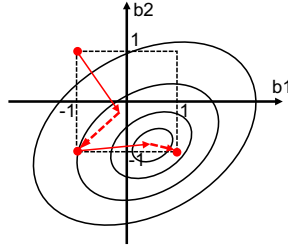

Figure 1: Suppose we only use two bits, b1 and b2, to encode the input. The circles represent the contour map of the loss function and the red line represents the update trajectory of the hash code, in which the full line denotes the Equation (2) while the dotted line denotes the Equation (3).

It is worth noting that our solution (3) is consistent with the conclusion of [27], in which the author has carried out a rigorous mathematical proof on the convergence of (3) (using theory from non-convex and non-smooth optimization [1, 3]). Different from their paper, we would pay more attention to the distinct meaning behind Equation (3) (the greedy choice), as well as the remarkable characters we will demonstrate below when combining (3) with neural network (notice that [27] has not used any deep learning method).

We believe that although (3) may not be the most effective method for solving discrete optimization problems, it is one of the best selections to combine with the neural network when confronting the discrete constraint. Reasons are listed as follows:

1) As is widely known that, neural network is updated by gradient descent which is also a greedy strategy as it updates the network toward the steepest descent direction of the loss function in each iteration, which demonstrates the high feasibility of using the greedy principle to handle optimization problem in neural network.

2) The similar update mode (calculate the gradient and then update the parameter) shared by Equation (3) and neural network has laid a solid foundation for combining them into an end-to-end training framework (see Section 2.2).

3) As is pointed out by [9], stochastic gradient descent (SGD) is equivalent to adding noise in the global gradient (calculated by the whole training samples), and appropriate noise in gradient not only bring regularization effect, but also helps the network to escape from some local minimum points and saddle points during optimization. From Figure 1 we can distinctly see that Equation (3) is exactly introducing "noise" into the original Equation (2) by the $sgn()$ function, thus using Equation (3) not only helps the network to handle the discrete constraint but also to some extent promotes the optimization process of the neural network.

Therefore, Equation (3) is a reasonable and effective way to solve the discrete hashing optimization in neural network, which will be further demonstrated with experiments later.

## 2.2 Back propagating the coding message with a new hash layer

In section 2.1 we have discussed why we choose (3) to deal with the discrete optimization in neural network, and in this section we would display how we implement (3) into the training system of the network with a newly designed hash layer.

Firstly variable $\mathbf{H}$ is introduced to split Equation (3) into:

$$\begin{cases} \mathbf{B}^{t+1} = sgn( \mathbf{H}^{t+1} ), & \text{(4a)} \\ \mathbf{H}^{t+1} = \mathbf{B}^t - lr * \dfrac{\partial L}{\partial \mathbf{B}^t}. & \text{(4b)} \end{cases}$$

Just recall that $\mathbf{H}$ denotes the output of the neural network while $\mathbf{B}$ denotes the hash code, and what we're going to do here is designing a new layer to connect $\mathbf{H}$ and $\mathbf{B}$ which should satisfy the Equations (4a) and (4b).

Knowing this, we could immediately find that what we need to do to implement Equation (4a) is simply using sign function in the forward propagation of the new hash layer, namely applying $\mathbf{B} = sgn(\mathbf{H})$ forward.

As for the Equation (4b), if we add a penalty term $\parallel \mathbf{H} - sgn(\mathbf{H}) \parallel_p^p$ (entrywise matrix norm) in the objective function to make it as close to zero as possible, then with Equation (4a) $\mathbf{B}^t = sgn(\mathbf{H}^t)$, we have:

$$
\begin{aligned}
\mathbf{H}^{t+1} &= \mathbf{H}^t - lr * \frac{\partial L}{\partial \mathbf{H}^t} \\
&= (\mathbf{H}^t - \mathbf{B}^t) + \mathbf{B}^t - lr * \frac{\partial L}{\partial \mathbf{H}^t} \\
&= (\mathbf{H}^t - sgn(\mathbf{H}^t)) + \mathbf{B}^t - lr * \frac{\partial L}{\partial \mathbf{H}^t} \\
&\approx \mathbf{B}^t - lr * \frac{\partial L}{\partial \mathbf{H}^t}.
\end{aligned}
\tag{5}
$$

Comparing (4b) with (5), we could finally implement Equation (4b) by setting:

$$
\frac{\partial L}{\partial \mathbf{H}^t} = \frac{\partial L}{\partial \mathbf{B}^t}
\tag{6}
$$

in the backward propagation of our new hash layer, which means the gradient of $\mathbf{B}$ is back transmitted to $\mathbf{H}$ intactly. Our method has been summarized in Algorithm 1.

---

**Algorithm 1** Greedy Hash

---

Prepare training set $\mathbf{X}$ and neural network $\mathcal{F}_\Theta$, in which $\Theta$ denotes parameters of the network.
**repeat**
  - $\mathbf{H} = \mathcal{F}_\Theta(\mathbf{X})$.
  - $\mathbf{B} = sgn(\mathbf{H})$ [forward propagation of our hash layer].
  - Calculate the loss function : $\mathcal{L}oss = L(\mathbf{B}) + \alpha \parallel \mathbf{H} - sgn(\mathbf{H}) \parallel_p^p$,
    where L can be any learning function such as the Cross Entropy Loss.
  - Calculate $\frac{\partial \mathcal{L}oss}{\partial \mathbf{B}} = \frac{\partial L}{\partial \mathbf{B}}$.
  - Set $\frac{\partial L}{\partial \mathbf{H}} = \frac{\partial L}{\partial \mathbf{B}}$ [backward propagation of our hash layer].
  - Calculate $\frac{\partial \mathcal{L}oss}{\partial \mathbf{H}} = \frac{\partial L}{\partial \mathbf{H}} + \alpha \frac{\partial \parallel \mathbf{H} - sgn(\mathbf{H}) \parallel_p^p}{\partial \mathbf{H}} = \frac{\partial L}{\partial \mathbf{B}} + \alpha\, p \parallel \mathbf{H} - sgn(\mathbf{H}) \parallel_{p-1}^{p-1}$.
  - Calculate $\frac{\partial \mathcal{L}oss}{\partial \Theta} = \frac{\partial \mathcal{L}oss}{\partial \mathbf{H}} \times \frac{\partial \mathbf{H}}{\partial \Theta}$.
  - Update the whole network's parameters.
**until** convergence.

---

## 2.3 Analyzing our algorithm's validity from a visual perspective

In this section, we would provide a new perspective to visualize and understand the two most critical parts in our algorithm:

$$
Forward: \quad \mathbf{B} = sgn(\,\mathbf{H}\,),
\tag{7}
$$

$$
Backward: \quad \frac{\partial L}{\partial \mathbf{H}} = \frac{\partial L}{\partial \mathbf{B}}.
\tag{8}
$$

Firstly, suppose there are two categories of input images. We set $\mathbf{H} = (h1, h2)$ and $\mathbf{B} = (b1, b2)$ (namely we only use two bits to encode the input image). As is shown in the forward part of Figure 2(a), sign function is trying to aggregate the data of each quadrant in H coordinate system into a single point in B coordinate system, and obviously learning to move the misclassified samples to the correct quadrant in H coordinate system is our ultimate training goal.

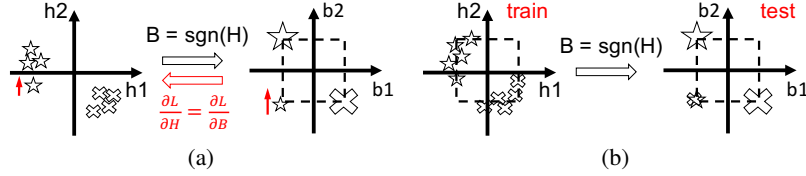

Figure 2: A visual perspective to observe (a) the aggregation effect of $sgn()$ and the back propagation from **B** to **H** in our algorithm (b) the quantization error generated by relaxation methods.

As is introduced earlier, most previous hashing methods relax $sgn()$ with tanh function or penalty term, whereas these relaxation schemes would produce a mass of continuous value violating the discrete constraint. The samples will not be imposed to aggregate strictly in the training stage (the left part of Figure 2(b)), while in the test phase the sign function is used to generate real binary code resulting in a different distribution from the training phase (the right part of Figure 2(b)). Certainly it will produce what we often call, the quantization error. In our proposed hash layer, sign function (Equation (7)) is directly applied in the training stage without any relaxation. The training samples will be strictly integrated in each quadrant, which enables our loss function ***foresee*** the error before the test phase and pay ***timely*** action on moving them.

Equation (8) that used in the back propagation of our hash layer, is the most significant part to settle the vanishing gradient problem of sign function. In order to propagate the moving information (update direction of each sample) from $\frac{\partial L}{\partial \mathbf{B}}$ to $\frac{\partial L}{\partial \mathbf{H}}$, we directly transmitted the gradient of **B** to **H** intactly, based on the principle that the aggregation effect of sign function does not change the quadrant of the input sample from **H** to **B**. As a consequence, direction that the misclassified samples need to move toward in the H coordinate system is exactly the direction learned by **B** in the B coordinate system (e.g., the red part in Figure 2(a)). Therefore Equation (8) enables **H** timely obtain the direction that the loss function expect **B** move toward, and it is (8) that helps our network to realize a fast and effective convergence.

By the way, it is noteworthy that even though the earlier researches on stochastic neurons [2, 24] have roughly mentioned about the straight through strategy (Equation (8)), our paper carefully study its derivation and demonstrate its performance in deep hashing coding domain. Moreover, we use (8) with the assistance of the penalty term $\| \mathbf{H} - sgn(\mathbf{H}) \|_p^p$ (has not seen in [2, 24]), which is nonnegligible to reduce the gradient bias through making **H** closer to **B** and improve the optimization property.

## 3 Experiments

We evaluate the efficacy of our proposed Greedy Hash in this section and the source code is available at: `https://github.com/ssppp/GreedyHash`.

### 3.1 Datasets

**CIFAR-10**  The CIFAR-10 dataset [14] consists of 60,000 32×32 color images in 10 classes. Following most deep hashing methods like [16] and [17], we have conducted two experiment settings for CIFAR-10. In the first one (denoted as CIFAR-10 (I)), 1000 images (100 images per class) are selected as the query set, and the remaining 59,000 images are used as database. Besides, 5,000 images (500 images per class) are randomly sampled from the database as the training set. As for the second one (denoted as CIFAR-10 (II)), 1,000 images per class (10,000 images in total) are selected as the test query set, and the remaining 50,000 images are used as the training set.

**ImageNet**  ImageNet [25] that consists of 1,000 classes is a benchmark image set for object category classification and detection in Large Scale Visual Recognition Challenge (ILSVRC). It contains over 1.2M images in the training set and 50K images in the validation set. Following the experiment setting in [4], we randomly select 100 categories, use all the images of these categories in the training set as the database, and use all the images in the validation set as the queries. Furthermore, 130 images per category are randomly selected from the database as the training points.

## 3.2 Implementation details

**Basic setting** Our model is implemented with Pytorch [23] framework. We set the batch size as 32 and use SGD as the optimizer with a weight decay of 0.0005 and a momentum of 0.9. For supervised experiments we use 0.001 as the initial learning rate while for unsupervised experiments we use 0.0001, and we divide both of them by 10 when the loss stop decreasing. In addition, we cross-validate the hyper-parameters $\alpha$ and $p$ in the penalty term $\alpha \parallel H - sgn(H) \parallel_p^p$, which are finally fixed with $p = 3$, $\alpha = 0.1 \times \frac{1}{N \cdot K}$ (using $\frac{1}{N \cdot K}$ term to remove the impacts of the various encoding length and input size) for CIFAR-10, while for ImageNet $\alpha = 1 \times \frac{1}{N \cdot K}$.

**Supervised setting** We choose the Cross Entropy loss as our supervised loss function $L$, which means we just apply softmax to classify the hash code $B \in \{-1, +1\}^{N \times K}$ without adding any retrieval loss (e.g., contrastive loss or triplet loss), and later we would display its outstanding retrieval performance despite merely using single softmax. Moreover, for fair comparison with the previous works, we use the pre-trained AlexNet [13] as our neural network, in which we append a new fc layer after fc7 to generate feature of desired length and then append our hash layer to produce binary code. We have compared our method with DSDH [16], HashNet [4], DPSH [17], DTSH [29], DHN [36], NINH [15], CNNH [30], VDSH [33], DRSCH [32], DSCH [32], DSRH [34], DPLM [27], SDH [26], KSH [21] under this supervised setup. It is worth noting that some of aforementioned methods such as DSDH have used the VGG-F [5] convolutional neural network, which composes of five convolutional layers and two fully connected layers the same as AlexNet (other methods including ours have selected), thus we consider the comparison among them is fair even though VGG-F performs slightly better on the classification of the original 1,000 classes ImageNet.

**Unsupervised setting** Inspired by [12], we choose to minimize the difference on the cosine distance relationships when the features are encoded from Euclidean space to Hamming space. Concretely we use $L = \parallel cos(h_1, h_2) - cos(b_1, b_2) \parallel_2^2$, in which $cos$ means the cosine distance, and $h$ means the feature in Euclidean space while $b$ means binary code in Hamming space. We use the pre-trained VGG16 [28] network following the setting in [6, 18], and we append a new fc layer as well as our hash layer to generate the binary code. We have compared with SAH [6], DeepBit [18], ITQ [8], KMH [10] and SPH [11] under this unsupervised setting.

## 3.3 Comparison on fast optimization

Firstly we compare our method with DSDH which can keep the discrete constraint during the whole optimization process just like ours. For fair comparison, we rerun the code released by the DSDH author and we follow the experiment setting in their program: using pre-trained CNN and encoding the images with maximum 48 bits as well as minimum 12 bits on supervised CIFAR-10 (I). The Mean Average Precision (MAP) during the training stage is shown on Figure 3(a).

We can see from Figure 3(a) that our method have achieved faster and better MAP promotion with both short and long bits (especially using the shorter one). In DSDH, the coding message from $\mathbf{B}$ is back propagated to the front network only by the loss term $\parallel \mathbf{B} - \mathbf{H} \parallel_2^2$, which will be deficient and unstable when the value is small, while our method uses $\frac{\partial L}{\partial \mathbf{H}} = \frac{\partial L}{\partial \mathbf{B}}$, capable of receiving the coding message rapidly and accurately as is analyzed in Section 2.3.

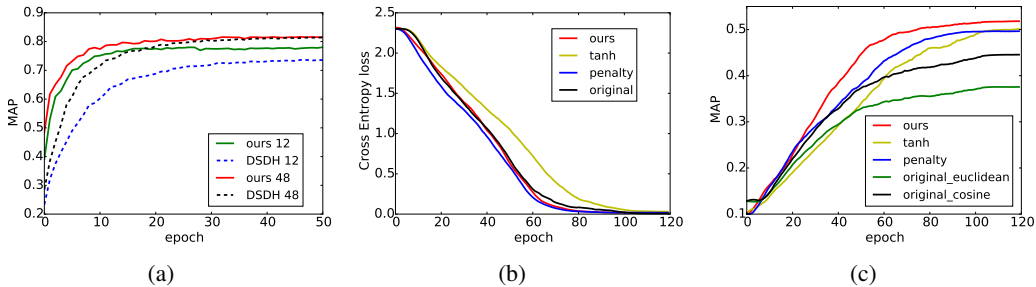

(a)　　　　　　　　(b)　　　　　　　　(c)

Figure 3: Fast optimization comparison with (a) DSDH and (b)(c) relaxation methods.

Next we compare our algorithm with the relaxation methods. We use 16 bits to encode the images and we train the AlexNet from scratch here to further demonstrate the learning ability. The dynamic classification loss and MAP are shown on Figure 3(b) and 3(c), in which label **tanh** means using tanh function as relaxation, **penalty** denotes method adding penalty term to generate features as discrete as possible, and **original** represents training without hash coding (the same length 16 but no longer limited to binary). We use two schemes to retrieve these original features, the Euclidean distance and the cosine distance.

As shown on Figure 3(b) and 3(c), our algorithm is one of the fastest methods to decrease the classification loss and simultaneously is the fastest and greatest one to promote the MAP owing to the better protection against quantization error. Among the results the tanh method is the slowest one probably due to the existence of saturation in tanh activation function. Moreover, it is interesting to discover that retrieval with hash binary code is better than the retrieval with original unrestricted features, probably because the softmax function generally needs adding retrieval loss to further constrain the image feature (for smaller intra-class distance and larger inter-class distance), which is exactly the merit of hashing that owns the nature of aggregating the inputs in each quadrant as is demonstrated in Section 2.3.

Thus our Greedy Hash is able to realize faster and better optimization in hash coding with CNN, and subsequently we will further demonstrate the accuracy of our generated hash code when compared with the state-of-the-art.

### 3.4 Comparison on accurate coding

**Supervised experiments**   Table 1 shows the MAP results on the supervised CIFAR-10 (I) dataset, in which we can explicitly see the better performance of our algorithm than both the state-of-the-art deep hashing methods and the traditional hashing with deep learned features as input ("SDH+CNN" denotes the SDH method with deep features). It is widely known that deep neural network needs more training samples to perform better, thus we conduct the second experiment on CIFAR-10 (i.e., CIFAR-10 (II)) whose training set is ten times the number of the first training setting. The MAP result is shown in Table 2. All the deep methods have made a big progress due to the augment of the training set, and even though the best result obtained by the previous work is as high as 93%, our method is capable of further improving it to 94%.

Notice that we have specially compared our method with DPLM [27] in Table 2, which distinctly reveals that using deep learned features as input can improve the performance of DPLM while our method can further boost the results as we better integrate hash coding with CNN and simultaneously realize efficient end-to-end training of the network.

Table 1: MAP on supervised CIFAR-10 (I), where "method+CNN" means traditional hashing methods with deep features as input.

| Method | Supervised | | | |
|---|---|---|---|---|
| | 12bits | 24bits | 32bits | 48bits |
| Ours | **0.774** | **0.795** | **0.810** | **0.822** |
| DSDH | 0.740 | 0.786 | 0.801 | 0.820 |
| DTSH | 0.710 | 0.750 | 0.765 | 0.774 |
| DPSH | 0.713 | 0.727 | 0.744 | 0.757 |
| NINH | 0.552 | 0.566 | 0.558 | 0.581 |
| CNNH | 0.439 | 0.511 | 0.509 | 0.522 |
| SDH+CNN | 0.478 | 0.557 | 0.584 | 0.592 |
| KSH+CNN | 0.488 | 0.539 | 0.548 | 0.563 |

Table 2: MAP on supervised CIFAR-10 (II).

| Method | Supervised | | | |
|---|---|---|---|---|
| | 16bits | 24bits | 32bits | 48bits |
| Ours | **0.942** | **0.943** | **0.943** | **0.944** |
| DSDH | 0.935 | 0.940 | 0.939 | 0.939 |
| DTSH | 0.915 | 0.923 | 0.925 | 0.926 |
| DPSH | 0.763 | 0.781 | 0.795 | 0.807 |
| VDSH | 0.845 | 0.848 | 0.844 | 0.845 |
| DRSCH | 0.615 | 0.622 | 0.629 | 0.631 |
| DSCH | 0.609 | 0.613 | 0.617 | 0.620 |
| DSRH | 0.608 | 0.611 | 0.617 | 0.618 |
| DPLM+CNN | 0.562 | 0.830 | 0.837 | 0.843 |
| DPLM | 0.465 | 0.614 | 0.643 | 0.671 |

Table 3 displays the retrieval result training on the supervised ImageNet, and our algorithm still have superior performance on this larger dataset. We should point out that the pairwise or triplet labels based methods generally need a high storage and computation cost to construct the input images group, which are infeasible for large-scale datasets. Our scheme learn hash code in a point-wise manner and consequently, it would be simpler and more promising for our method to apply in the practical retrieval system.

Table 3: MAP@1000 on supervised ImageNet.

| Method | Supervised | | | |
|---|---|---|---|---|
| | 16bits | 32bits | 48bits | 64bits |
| Ours | **0.625** | **0.662** | **0.682** | **0.688** |
| HashNet | 0.506 | 0.631 | 0.663 | 0.684 |
| DHN | 0.311 | 0.472 | 0.542 | 0.573 |
| NINH | 0.290 | 0.461 | 0.530 | 0.565 |
| CNNH | 0.281 | 0.450 | 0.525 | 0.554 |
| SDH+CNN | 0.298 | 0.455 | 0.554 | 0.585 |
| KSH+CNN | 0.160 | 0.298 | 0.342 | 0.394 |

Table 4: MAP@1000 on unsupervised CIFAR-10 (II).

| Method | Unsupervised | | |
|---|---|---|---|
| | 16bits | 32bits | 64bits |
| Ours | **0.448** | **0.472** | **0.501** |
| SAH | 0.418 | 0.456 | 0.474 |
| DeepBit | 0.194 | 0.249 | 0.277 |
| ITQ+CNN | 0.385 | 0.414 | 0.442 |
| KMH+CNN | 0.360 | 0.382 | 0.401 |
| SPH+CNN | 0.302 | 0.356 | 0.392 |

**Unsupervised experiment** The MAP@1000 result on unsupervised CIFAR-10 (II) is shown on Table 4 (again, "ITQ+CNN" denotes the ITQ method with deep features). Our method is still able to improve the performance as before in spite of our rough unsupervised objective function. In view of the improvement by our algorithm in both the supervised and unsupervised tasks, we demonstrate that with small modifications to the original network, our method can easily transfer to various occasions that need hash coding.

### 3.5 Coding with shorter length

From all the retrieval results above, it is interesting to discover that the shorter encoding bits we use, the larger margin we obtain when comparing with the other methods. It seems that our method can maintain a decent retrieval performance even with restricted encoding length. Thus we conduct the following experiment on supervised CIFAR-10 (I) using different encoding length shorter than the common minimum 12 bits (but larger than 4 bits as there are 10 classes in CIFAR-10) to further seek more outstanding character of our method.

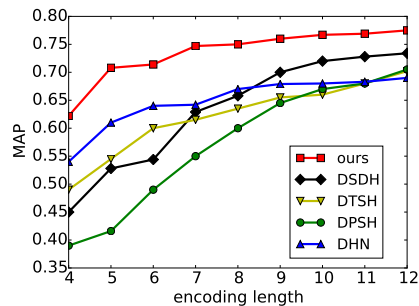

Figure 4: Retrieval experiments with shorter encoding length (4-12 bits).

The result is shown on Figure 4, from which it is impressive to find that our proposed scheme indeed produce nice results with shorter hash code. With this charming nature, our method can process the large-scale image set with higher storage efficiency and faster retrieval speed, which we think is extremely useful in practical application.

## 4 Conclusion

In this paper we propose to adopt greedy algorithm to tackle the discrete hashing optimization, and we design a new neural layer to implement our approach, which fixedly uses the sign function in forward propagation without any relaxation to avoid the quantization error, while in backward propagation the gradients are transmitted intactly to the front layer, preventing the vanishing gradients and helping to achieve fast convergence. The superiority of our method is proved from a novel visual perspective as well as abundant experiments on different retrieval tasks.

## Acknowledgments

This work was supported in part by the National Key R&D Program of China under Grant 2017YF-B1002400, the National Natural Science Foundation of China under Grant 61671027, U1611461 and the National Key Basic Research Program of China under Grant 2015CB352303.

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
