[Supplementary Material]

# Greedy Hash: Towards Fast Optimization for Accurate Hash Coding in CNN (Supplementary Material)

## 1   Visualization of feature distribution in H coordinate system

In Section 2.3 of our main paper, we analyse our method's rationality and validity from a visual perspective, and in this section we would like to display the feature distribution of the real image dataset in the H coordinate system (before using activation function) to further demonstrate what we declare in Section 2.3 . We use two bits to encode the input image (as we desire to display them on 2D figure) and select four classes of the CIFAR-10 [1] (as two binary bits maximally represent four classes) to construct this experiment. Besides, we use the CIFAR-10 (I) setup (100 images per class are selected as the query set and the remaining 5900 images per class as database, from which 500 images per class are randomly sampled as training set) as well as the supervised experiment setting described in the main paper.

The feature distribution of query set is shown on Figure 1, in which **tanh** means using tanh function as relaxation, **penalty** denotes method skipping the activation layer but adding a penalty term to generate features as discrete as possible, **original** represents training without coding, which uses the unrestricted features (the same length of hash code but no longer limited to binary) for classification and retrieval.

Figure 1 strongly proves that the feature distribution in H coordinate system is exactly what we describe and exhibit in Section 2.3 of the main paper, and we can see from the distribution that our method performs better to separate the samples with different labels apart as far as possible, while other methods have more values around zero which will obviously introduce more quantization error than ours. Moreover, the retrieval MAP of each method on this four-classes CIFAR are: 0.87 for our proposed hash layer, 0.85 for **penalty** and 0.84 for **tanh**, quantitatively demonstrate the superiority of our method.

## 2   Supplementary experiment on CIFAR-100

The CIFAR-100 dataset [1] is just like the CIFAR-10 except it has 100 classes, in which there are 500 training images and 100 testing images per class. For this dataset, we simply use the 10,000 images in test set as the query set, and the remaining 50,000 images are used as training set.

Here we use CIFAR-100 to supplement an experiment similar to Section 3.3 of the main paper to further analyze our method's efficiency on a harder dataset. The dynamic classification loss and Mean Average Precision (MAP) during the training stage are shown on Figure 2(a) and 2(b). Moreover, we still use 16 bits to encode the images and we train the AlexNet from scratch. Label **tanh** means using tanh function as relaxation, **penalty** denotes method adding a penalty term to generate features as discrete as possible, **original** represents training without coding (again we use two schemes to retrieve these original features, the Euclidean distance and the cosine distance).

From Figure 2 we can clearly see that, our method is capable of decreasing the classification loss rapidly and simultaneously is the fastest and greatest one to promote the MAP retrieval performance

(a) ours (b) tanh

(c) penalty (d) original

Figure 1: Feature distribution in the H coordinate system.

(a) (b)

Figure 2: Experiment on CIFAR-100 to display (a) cross entropy loss of classification and (b) MAP for retrieval during the training epochs.

on this harder dataset, which again demonstrates that our method can achieve fast optimization for accurate hash coding in CNN.

## References

[1]  A. Krizhevsky and G. Hinton. Learning multiple layers of features from tiny images. 2009.