[Reviews · NeurIPS 2018]

Reviewer 1



The paper introduces a greedy approach to tackle the NP-hard problem of hashing with deep learning. Although the main idea of “Discrete Proximal Linearized Minimization” is borrowed from “A Fast Optimization Method for General Binary Code Learning”, the paper shows how to use this approach in combination with neural networks and SGD. Also, the reported results beat state of art methods. About the experimental setup, there is no comparison with other methods on widely used datasets such as NUS-WIDE, which has an important distinction that samples can have multiple labels. It would be very good to see a comparison against such state of art methods as DVSQ (Deep visual-semantic quantization for efficient image retrieval) and DMDH (Deep hashing via discrepancy minimization). Post-rebuttal comments: After reading the rebuttal and all the reviews, the Authors have addressed convincingly my concerns. They have provided results for additional experiments and have shown that their method also works under a multilabel setup. The third reviewer pointed out that similar ideas were proposed for the case of stochastic neurons. However, this paper still is the first one that applies these ideas to hashing. Although what pointed out by the third reviewer diminishes the novelty aspect of the paper, I have not changed my initial overall score.

Reviewer 2



The paper deals with the problem of deep hashing, namely learning a deep CNN to produce binary codes at its output, with application to retrieval with nearest neighbor search and other tasks. The main challenge in this formulation is the discrete nature of the target output hash codes. Their non-differentiable nature prevents the use of vanilla SGD for training. There are various approaches in the literature to circumvent this issue, such as using a tanh(beta.x) function with annealed beta or adding a penalty to encourage the codes to be near-binary. The approach advocated by the paper is to adopt a pass-through approximation for the backward step: One simply uses the pointwise sign(x) function in the forward step (both during training and inference), and simply leaves the gradient intact during backward propagation, as if the sign(x) function was not there, which have also been advocated by other deep learning formulations that employ activation discretization. The authors evaluate their approach roughly following the experimental protocols of the related DSDH [15] approach, showing reasonable improvements over [15] and other baseline methods. Pros: Simple and effective method, reasonable improvements in performance and training speed compared to the closest DSDH baseline. Cons: Relatively small novelty. Discussion of the proposed approach in Sec. 2.2 and 2.3 is too lengthy.

Reviewer 3



The paper presents a greedy approach to train a deep neural network to directly produce binary codes that build on the straight through estimator. During forward propagation the model uses the sgn output whereas at the back-propagation stage it passes derivatives as it the output were a simple linear function. There are relevant papers that already proposed such an approach and that are not referred to as earlier work: [1] Estimating or Propagating Gradients Through Stochastic Neurons for Conditional Computation, Bengio Y. et al. [2] Techniques for learning binary stochastic feedforward neural networks, Tapani R. et al. The experimental setting is not very clear and I would suggest the authors to better explain the supervised setting. Do they produce a binary code of length k and then classify it with a single final output layer? If so then I would like to see an ablation study on this. What is the relative contribution of using this loss vs the contrastive or triplet loss that is used in many of the other approaches? What is the influence of the pre-trained network? What would be the performance if the method were to be trained from scratch end-to-end? This would allow for a better assessment of the greedy relaxation proposed here proposed for binarizing the output layer. Regarding other works that have tackled the problem of hashing with neural networks I see missing: [3] Multimodal similarity-preserving hashing, Masci J. et al. [4] Semantic Hashing, Salakhutdinov R. et al The comparison in 3.5 claims a performance margin that is larger at short code lengths wrt other methods, however I see only one of the algorithms in the figure. Please report more. Overall an interesting application paper that is clearly written, but that should include further references and readjust the contribution section considering that the straight through is not novel. ============================================================== After having read the rebuttal I decided to increase my score as the authors addressed all my concerns.